# A Wireless Autonomous Real-Time Underwater Acoustic Positioning System

**DOI:** 10.3390/s22218208

**Published:** 2022-10-26

**Authors:** François-Marie Manicacci, Johann Mourier, Chabi Babatounde, Jessica Garcia, Mickaël Broutta, Jean-Sébastien Gualtieri, Antoine Aiello

**Affiliations:** 1UMS 3514 Plateforme Marine Stella Mare, Université de Corse Pasquale Paoli, 20620 Biguglia, France; 2Société Informatique et Technologique Corse (S.I.T.E.C), Parc Technologique d’Erbajolo, 20601 Furiani, France

**Keywords:** acoustic positioning telemetry system, real time, tracking

## Abstract

Recent acoustic telemetry positioning systems are able to reconstruct the positions and trajectories of organisms at a scale of a few centimeters to a few meters. However, they present several logistical constraints including receiver maintenance, calibration procedures and limited access to real-time data. We present here a novel, easy-to-deploy, energy self-sufficient underwater positioning system based on the time difference of arrival (TDOA) algorithm and the Global System for Mobile (GSM) communication technology, capable of locating tagged marine organisms in real time. We provide an illustration of the application of this system with empirical examples using continuous and coded tags in fish and benthic invertebrates. In situ experimental tests of the operational system demonstrated similar performances to currently available acoustic positioning systems, with a global positioning error of 7.13 ± 5.80 m (mean ± SD) and one-third of the pings can be localized within 278 m of the farthest buoy. Despite some required improvements, this prototype is designed to be autonomous and can be deployed from the surface in various environments (rivers, lakes, and oceans). It was proven to be useful to monitor a wide variety of species (benthic and pelagic) in real time. Its real-time property can be used to rapidly detect system failure, optimize deployment design, or for ecological or conservation applications.

## 1. Introduction

Human activities and climate changes are rapidly influencing marine environments and, consequently, the spatial and temporal distribution of marine organisms. In this context it has thus become essential to develop tools and technologies able to monitor biodiversity and animal behavior in real-time for conservation and management purposes.

Acoustic telemetry offers a robust tool to monitor the behavior and ecology of aquatic animals. Each new generation makes it possible to track more individuals at finer spatio-temporal scales, over longer durations, while covering larger areas, in environments that were once inaccessible, and using smaller acoustic transmitters [1,2,3,4].

Different high resolution acoustic positioning systems have been developed and proposed by telemetry manufacturers. To the best of our knowledge, none of them have the following characteristics: simple to deploy, maintain and retrieve; deployable in various environments (rivers, lakes and ocean); usable for long periods of time; energy self-sufficient; high resolution; allow real-time access to data on multiple devices (mobile, tablet and computer); use free and flexible position reconstruction software.

While active tracking can yield fine-scale movement information, it is only limited to a few individuals over a short period of time [5]. Classical passive monitoring is used to remotely quantify longer-term presence–absence and movements of multiple individuals over larger temporal and spatial scales. If it is less time-consuming and labor-intensive, it lacks the high-resolution movement data and positioning accuracy from active tracking studies. Their deployment is generally time-consuming and exhausting as it requires the presence of divers for long periods of time to fix all the receivers constituting the mesh on the seabed. Moreover, fault detection cannot be done remotely, which could compromise the experiments, and the retrieval of receivers has the same constraints as their deployment.

Vemco (now Innovasea Systems Inc., Boston, MA, USA) developed the VR2W Positioning System (VPS) composed of acoustic monitoring receivers and fixed synchronizing transmitters (“sync tags”) providing high-resolution positioning with well-established performance [3]. The VPS can cover large areas but does not offer real-time positioning. Lotek recently introduced the Juvenile Salmon Acoustic Telemetry System (JSATS), a high-frequency positioning system that can simultaneously track more than 100 tagged fish with unprecedented spatial and temporal resolution, revealing the most realistic trajectory of the animal’s movements [4]. However, this system requires the deployment of a high density of receivers due to limited detection ranges in the marine environment when using high frequencies. In addition, both systems still suffer from the need to retrieve the receivers and use proprietary software to download and access the data weeks to months after they were recorded. This may also result in data loss if damage to the system during deployment is not detected.

Real-time data access to acoustic receivers has been offered by Vemco (Innovasea, Boston, WA, USA), Lotek (Newmarket, ON, Canada) and HTI (Seattle, WA, USA) manufacturers. Vemco Radio Acoustic Positioning (VRAP) is a wireless system using surface receiving stations (buoys) that rely on an FM radio signal to synchronize the clocks of the receivers and relay the acoustic signal to a computer on land for integration and positioning. Unfortunately, the cost and battery life of acoustic radio relays have limited its widespread application [6]. Lotek Wireless MAP600 and HTI Model 290/291 provide an alternative approach in which hydrophones are directly connected to a shore-based receiving system using cables [7], thus solving the energy constraint of the VRAP system. However, these systems are geographically restricted to small areas (e.g., river or lake systems), as the interface needs to be installed on shore or fixed to sea installations. The Internet of Fish (IoF) concept merges acoustic fish telemetry with the LPWAN-based LoRa communication protocol, a long-range low-power technology, allowing real-time access to telemetry data for fish farm monitoring [8]. Despite the real-time availability of data, the gateways must be fixed and powered on shore or sea-based facilities. In addition, the duty cycle imposed by the legislation limits the bandwidth and sets the maximum payload size at 135 bytes with a spreading factor of 9. This limits the size of the mesh and/or the frequency of data collection.

We propose a new high-resolution, affordable (around USD 5000 per unit in June 2022), real-time acoustic tracking system designed for long-term tracking as it is completely self-sufficient in terms of energy due to its solar panel. The positioning is based by default on trilateration using the time difference of arrival (TDOA) algorithm but users are offered the possibility to choose another positioning method such as Yet Another Positioning Solver (YAPS) [9] as the raw data are freely available. Moreover, this system was designed to be easily deployed and retrieved by a two-person crew without the need for divers. Due to the real-time analysis of the data and our responsive web supervision interface (mobile, tablet and computer), failure detection is possible from the very beginning. In this paper, we describe and test the performance of our system and illustrate its application using small-scale monitoring of marine organisms at sea. Finally, we discuss its application and future developments.

## 2. Materials and Methods

### 2.1. Physical Characteristics

Acoustic receivers are embedded inside custom buoy devices (Figure 1, step 1) derived from a B & W International Type 6500 case, which is IP67 certified and made of polypropylene (Figure 2). A solar panel is integrated at the top and is protected by a plexiglass plate. The outlet of the hydrophone is placed on this same side and its sealing is ensured by a cable gland. The attachment system is a stainless-steel hoop encompassing the suitcase with a ring located on the underside allowing the buoy to be tethered to its mooring. Inside, the electronic card and the two batteries are held in place by foam. The physical characteristics and shape (52 × 70 × 30 cm and 16.8 kg) of the acoustic receivers facilitate its transport, deployment and maintenance.

### 2.2. Embedded System

The computer board is responsible for detecting and dating the acoustic signals emitted by the transmitters, and is encased in a waterproof box, connected to its peripherals (batteries, solar panel, light, hydrophone and antennas) via waterproof electrical outlets (Figure 3). It has been custom-designed to meet our needs in terms of filtering, computing power, power consumption and peripherals. The board includes an STM32F446ZET6 SoC based on an ARM Cortex-M4 architecture with a frequency of 180 MHz, 512 KB of programmable flash memory and 128 KB of RAM. A RTOS was used for low-level resource management and our software is written entirely in C.

An 8-level acoustic filter passively filters the original signal (range [60–90 kHz]) and sends it to the SoC via the DMA module to transform the acoustic signal from the tags into a detection. Date and GPS location of the receiver are then associated and periodically sent via GSM to servers (Figure 1, step 2) to be stored and processed later (Figure 1, step 3).

Power supply management is performed via the Maximum Power Point Tracking (MPPT) module present on the board, which is connected to two Dryfit Gel A512/10S batteries from Sonnenschein (12 V, 10 Ah) and a 35 Watts (Watt-peak) solar panel from Phaesun. The solar panel was chosen to make our device autonomous in terms of energy throughout the year with sunshine conditions encountered in the Mediterranean Sea. In addition, the batteries were chosen to ensure up to two weeks of autonomy under normal conditions of use in case of solar panel failure, which we consider sufficient to schedule a maintenance intervention. The average power consumption of the card was measured at around 650 mW with a maximum detected at 1 W in the case of permanent data sending. From this information, we deduced a minimum autonomy of 240 h (worst case) and an average autonomy of 369 h (average case) with a total battery capacity of 20 Ah. In addition, the theoretical autonomy of our acoustic receiver was confirmed by the freely available online-tool of the European Union Photovoltaic Geographical Information System (PVGIS), available online: https://re.jrc.ec.europa.eu/pvg_tools/en/ (accessed on 7 August 2022). The parameters used are shown in Figure 4. Note that the consumption data were modified to reflect uniform consumption throughout the day. The results (see the report generated by PVGIS (Appendix A)) show that, in theory, the batteries will never be completely discharged with a probability of 98.99% that the charge will stay between 92% and 100%.

### 2.3. Server and Processing

The stored raw detections sent from the board are analyzed periodically (every two minutes) and used to reconstruct the position of acoustic emitters via a time difference of arrival (TDOA) algorithm [10].

The time difference of arrival (TDOA) algorithm is one of the most popular algorithms for finding the 2D/3D location of a moving target. It requires the receivers to be highly synchronized and is based on when the transmitted signal was received by each beacon. In our case, each receiver uses its GPS clock to synchronize with the others. For each pair of receivers, the difference in the distance between each receiver and the target (Δd) is computed from the difference of arrival time Δt and the speed of the signal in the medium c as:Δd=c∗Δt

Using another identity based on the location of each receiver $\left(x_{i},\;y_{i},\;z_{i}\right)$ and the unknow location of the target $\left(x,\;y,\;z\right)$, Δd can also be expressed as:$$\Delta d={\left[{\left(x_{1}-x\right)}^{2}+{\left(y_{1}-y\right)}^{2}+{\left(z_{1}-z\right)}^{2}\right]}^{1/2}-{\left[{\left(x_{2}-x\right)}^{2}+{\left(y_{2}-y\right)}^{2}+{\left(z_{2}-z\right)}^{2}\right]}^{1/2}$$

The last equation can be converted into a hyperbola and used to determine the unknown location of the target when enough equations are available, meaning enough receivers have detected the signal (three receivers for 2D positioning and four for 3D positioning), by finding their intersection [11].

In addition, decoding algorithms are applied to extract the information (ID, depth, acceleration, etc.) encapsulated in the coded signals. All results are stored in our database and are ready to be viewed in our responsive (smartphone, tablet, laptop) Web interface (Figure 1, step 4, and Figure 5). This interface is modular and additional features can be added as required. For example, a zone-crossing service has been implemented, alerting users by email when an individual crosses a previously defined polygon. Finally, the IT administration can also be carried out from our WEB application. A video presentation of our interface can be found in the Appendix A.

### 2.4. Installation Procedure

The installation of the system does not require diving and receivers can be installed from the surface by two operators. A preliminary step consisting of range tests is necessary to determine the optimal distance between receivers [12]. We recommend an equilateral triangular grid-shape at which 75% acoustic signals are recorded by the receivers. Obviously other grid designs can be explored (see [4]). On site, the moorings must first be placed on the seabed at predefined GPS locations corresponding to the nodes of the grid. A signaling buoy is then attached to it on which the receiver device is also secured (Figure 6). A final step consists in hooking a weight of ~2 kg to the hydrophone to ensure its maintenance at the desired depth.

The length of the rope connecting the mooring to the surface buoy should be equal to the depth of the seabed, approximately. The receiver attached to this rope should be placed at between three and five meters to ensure it does not become submerged in rough sea conditions. This could result in excessive force on the moorings and cause them to move, changing the shape of the grid and thus the performance of the system. The maximum depth of operation and its impact on performance have not yet been investigated. The maximum depth reached was 50 m during our experiments with lobsters.

### 2.5. Performance Tests

On 16 September 2019, an experiment was conducted in the Mediterranean Sea, about 10 km south of Bastia in Corsica (Figure 7), to determine the positioning accuracy and the maximum operating distance of our system using three receivers and a pinger (P-HP13 70 kHz 3 s, Thelma Biotel, Trondheim, Norway).

Preliminary range tests resulted in a conservative distance between buoys of 270 m (75% of detections). A towed positioning test was carried out by attaching one transmitter to a weighted line that was hung on one side of the boat keeping the transmitter at depth between 13 and 16 m. The boat slowly drifted within the array at a mean speed of 9.2 km h^−1^ (Figure 8) in two steps: (1) randomly moving between the devices to evaluate the accuracy of the positioning system within the array; and (2) moving away from the center of the array to determine the maximum operating distance of our system. The position of the boat tracks during the towed test was recorded using a regular portable GPS (Garmin GPS73, accuracy: ±3–5 m). The reference positions of the boat were then compared to those inferred from our positioning system.

As the GPS could not be synchronized with the transmitters, we used three different approaches to measure the accuracy of our system: the difference in space, time, and both space and time. The first (space-based) method selects, for each calculated position, the closest GPS point in a time interval of 30 s. As there are intersections in the path of the boat, this constraint was applied to prevent the selection of points that do not belong to the correct segment. The second (time-based) method simply selects the closest GPS point in time. We propose a third (space- and time-based) method that selects the GPS point that minimizes the Euclidean distance in a 2D space composed of the difference in space between the points on the x-axis and the difference in time on the y-axis.

In addition, the reconstruction rate (the ratio of the number of calculated points to the number of expected points) as a function of the distance from the furthest receiver was also calculated. In the ideal case, one point is expected every three seconds along the path due to the fixed period of the acoustic transmitter (pinger). To calculate the reconstruction rate, a high-resolution path was constructed by interpolating new points between existing points. This high-resolution version was then divided into 21 s segments (a period that guarantees a ratio of less than or equal to one due to noise and uncertainty) and the number of expected points was compared to those found after reconstruction.

The three methods gave contrasting results. The space-based method gave the best accuracy at the cost of the highest average time difference, even with a time constraint of 30 s applied during the selection process. Conversely, the time-based method gave the poorest results despite the lowest average time difference, as shown in Figure 9. The third method represents a good compromise, and its results are used here for comparison with the performance of other systems found in the literature.

### 2.6. Study Cases

To illustrate the utility and efficiency of our system, we applied it to free-living tagged organisms. The Stella Mare research platform of the University of Corsica has focused on the monitoring of marine animals [5] for the ecological restoration of the Corsican coast. In this context, a common dentex (*Dentex dentex*) was caught in Nonza on the west side of Cap Corse (Corsica), equipped with a coded acoustic transmitter (ID-LP13-69 kHz Thelma Biotel (Trondheim, Norway), random delay: 30–90 s) and released at its capture location as part of the MoPamfish project. Similarly, a common spider crab (*Maja squinado*) was equipped with a similar coded transmitter (ID-LP13-69 kHz Thelma Biotel, random delay: 30–90 s) that was attached to one of its legs and was released in the same area in July 2020. In June 2021, a European lobster (*Homarus gammarus*) was equipped with a continuous pinger (P-HP13 67 kHz 3 s, Thelma Biotel) on its claw and released at a wreck on the east side of Cap Corse.

A network of 11 acoustic positioning devices recorded the movements of the common dentex and the spider crab in Nonza and four devices deployed around the wreck recorded the movements of the lobster. In each case, positions were transmitted to the server and visualized on the Web interface in real-time.

After each experiment, we downloaded the data and applied a filtering procedure by removing all unrealistic positions (potential false detections) that were located more than 200 m from at least one of the receivers. We then fitted a continuous time-correlated random walk (CTCRW) movement model [13] to the data, which predicts temporally regular positions (i.e., every minute for coded tags and every three seconds for the pinger). The CTCRW model was fitted using the crawl package for R [14], setting the positioning error to 10 m, the maximum mean value that was extracted from the performance tests (see below).

## 3. Results

The overall accuracy of our system is 7.13 ± 5.80 m (mean ± SD). Inside and outside the mesh, the accuracy is 6.04 ± 3.94 m and 12.68 ± 9.59 m, respectively. We also evaluated the position reconstruction rate and found that more than one-third of signal positions were reconstructed within about 278.5 m from the furthest receiver position (Figure 10).

On 4–16 June 2021, 48,089 raw positions of the lobster were reconstructed by our system, of which 100% were within 200 m of some receiver (yellow track in Figure 11A). Between June 23 and September 10, 1225 positions of the dentex were recorded (Figure 11B). Initially, filtering to remove outliers and erroneous positions located more than 200 m from at least one of the devices yielded 834 realistic positions (68%). For the spider crab (Figure 11C), 375 positions were determined, and the filtering allowed to remove two outliers (<1% of all positions). The observed difference in false detections between the dentex and the spider crab might be due to the fact that the fish is more mobile than the crustacean. A CTCRW model was applied to the filtered data for each tagged animal to reconstruct a realistic path (Figure 10).

## 4. Discussion

To position our system in the context of commercially available products, the performances of eight acoustic telemetry positioning systems from nine studies were compiled (Appendix A). Since all our results were obtained for mobile transmitters, we limit our comparison to the performance of systems whose transmitters were also mobile.

Within the mesh, our system performs better than the HR-VPS (Innovasea, Boston, MA, USA) [15], with a median positioning error of 5.4 m compared to the 8.3 m of the Innovasea system. In another case using VPS-VR2W (Innovasea), the median error of 4.07 m was closer to the median error of our system [3]. Our average positioning error of 6.04 m is also comparable to the average 5.3 m positioning error of the WHS 3050 MAP 200 kHz (Wireless) from Lotek (Newmarket, ON, Canada) [9]. Outside the grid, our system is slightly limited with a median error of 10.97 m compared to the median error of 4.12 m with the VPS-VR2W (Innovasea, Boston, MA, USA), corresponding to a 6.85 m difference in positioning error. Nonetheless, it is still demonstrating encouraging performances, with an average positioning difference of 2.13 m compared to the VPS-VR2W, whose results are from [16].

The JSTATS system (Lotek WHS 4250-L 416.7 kHz receivers and Lotek L-AMT coded transmitters, 416.7 kHz, Newmarket, ON, Canada), although operating at a higher frequency (416.7 kHz), offers an interesting performance. Even if the accuracy of the positions (median 10.4 m) is lower than that of our system (median 5.83 m), this system shows great resistance to signal collisions (simultaneously able to track >170 individuals) and offers the possibility to monitor the movement of individuals on a very fine spatio-temporal scale [4]. However, the detection range of the JSTATS is reduced in marine conditions, which would require installing more receivers to cover the same area.

While the system presented here is not a more effective alternative than the commercial acoustic positioning systems currently available, the accuracy inside the mesh (6.04 ± 3.94 m) still represents a good level of performance, allowing movements to be studied at relatively high resolution. It does offer several additional features that may be of great interest to researchers and managers involved in monitoring studies of marine organisms, and responds to the previous suggestions [2]. In fact, it is the only product offering a real-time positioning system that is easily and rapidly deployable. Obviously, some improvements are still necessary; in particular, energy autonomy is not completely achieved.

Even if our energy management system has been validated by theory with the sunshine conditions encountered in the Mediterranean, our receivers have been confronted with the harsh reality of the field. As shown in Figure 12, our receivers did not last more than two weeks when deployed in winter. The experimental site was located near steep mountains, which considerably reduced the solar energy available during the day (the sun appearing in late morning over the mountain). Moreover, we suspect that the plexiglass plate is responsible for a decrease in the efficiency of the solar panel. The is particularly likely because its degradation by the difficult conditions encountered at sea led to an increase in its opacity over time and after multiple experiments.

Real-time communication, processing and visualization ensure continuous availability of data, as reconstructed positions are available on the Web interface, at a maximum of two minutes after the emission of the sound signals under normal conditions. This represents a major advantage since it allows preliminary analyses to be conducted, and alerts researchers and stakeholders about any unusual behavior without waiting six months or more to retrieve the receivers and download the data, as it is common practice with traditional acoustic receivers. In addition, it allows monitoring the status and functioning of the material. It is therefore no longer necessary to maintain the receivers during dedicated dives to ensure the operational continuity of the equipment. Receiver failures can be identified allowing rapid on-site intervention, thus limiting the negative impact of any interruption in data collection. This real-time information also allows the design of the array to be adjusted if tagged individuals leave the study area or move to the edges. This is made possible because the system has been designed to be easily deployable and redeployable. Unlike most acoustic receivers on the market, which involve the assistance of divers, all operations for the presented system can be carried out by two people in a short time (~10 min on average per device) from a boat.

The Lotek WHS 600 MAP 200 kHz Cabled (Newmarket, ON, Canada) [7] and the recent IoF system [8] also offer real-time data access using wired and LPWAN based solutions, respectively. Cables in marine conditions are a constraint that limits the application of the first solution in marine contexts, whereas the second solution requires a fixed and powered gateway, and is therefore more suitable for real-time monitoring in marine farms or other marine infrastructures. Our system is composed of independent units that can be deployed anywhere without the need to deploy a local LoRa network as it is based on the widely used GSM network.

Another benefit is that our system has been positively tested using both coded and continuous acoustic transmitters. Coded tags can integrate sensors allowing real-time visualization of individual parameters (depth, acceleration, temperature, swimming speed, etc.) while continuous tags with different frequencies and short transmission intervals provide our product with more flexibility.

Finally, our experimental tests showed that it can perform well on benthic and semi-pelagic species, making it a flexible tool for a reasonable price below USD 5000 (estimated in June 2022).

Future work will focus on improving the accuracy, energy efficiency and robustness of our system, and particularly the composite materials for long-term exposure to harsh marine conditions. The web interface will be redesigned to incorporate real-time analysis capabilities, as well as a privacy option to restrict access to real-time data to protect the real-time sensitive positions of endangered species.

## Figures and Tables

**Figure 1 sensors-22-08208-f001:**
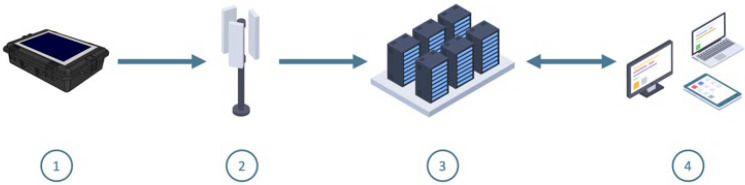
Schematic diagram of the system subdivided in four parts numbered from 1 to 4 describing the transmission of data from acquisition to delivery: (1) in situ recording devices, (2) communication in real time via GSM network, (3) raw data storage and processing server and (4) export and visualization tools.

**Figure 2 sensors-22-08208-f002:**
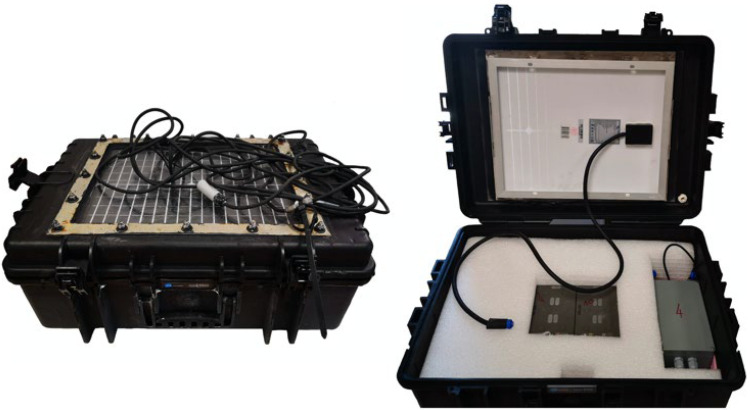
On the left, the external appearance of the acoustic receiver presenting its solar panel on the upper side. This case was chosen for its practicality, weight and shape for easy storage and handling before, during and after the experiments. On the right, the internal appearance of the acoustic receiver showing the electronic board enclosed in its waterproof case and the two batteries. Each component is enclosed in a polystyrene panel to prevent damage in rough sea conditions.

**Figure 3 sensors-22-08208-f003:**
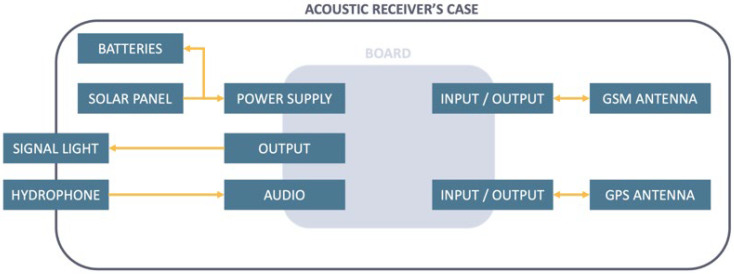
Diagram of the different elements integrated in an acoustic receiver. The electronic board, located in the center in grey, is enclosed in a waterproof case and linked to its peripherals by waterproof connectors. It is responsible for processing the signal from an acoustic transmitter and sending the data to the server. The solar panel, batteries and both GPS and GSM antennas are located inside the acoustic receiver case, while the hydrophone and the signal light are located outside.

**Figure 4 sensors-22-08208-f004:**
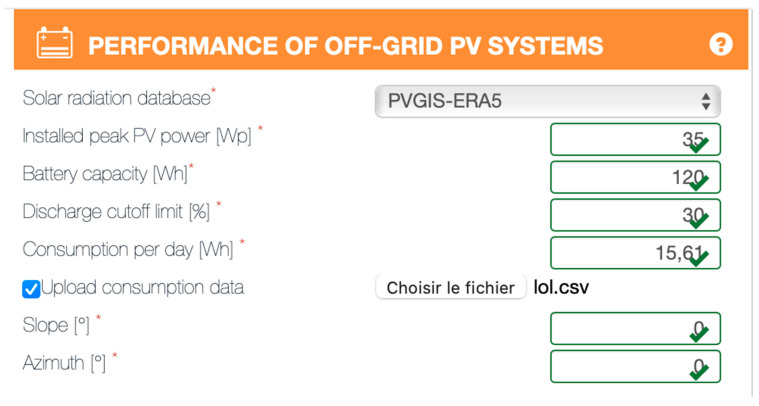
Screenshot of the parameters in the PVGIS online tool used to confirm theoretically the choice of our solar panel. The symbols * correspond to the required parameters to input.

**Figure 5 sensors-22-08208-f005:**
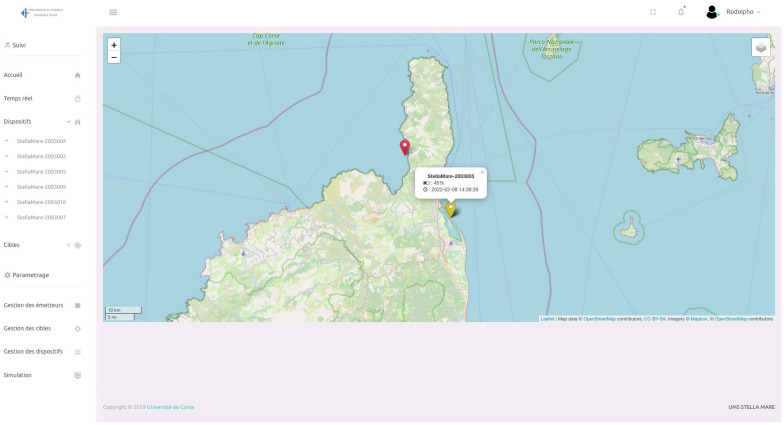
Screenshot of the main view of the monitoring interface. It provides a cartographic display as well as a sidebar displaying the status of receivers and transmitter IDs.

**Figure 6 sensors-22-08208-f006:**
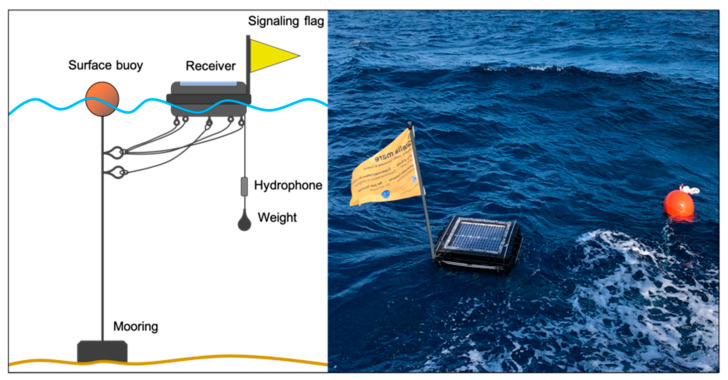
On the left, an illustrative diagram of the deployment of an acoustic receiver at sea. It shows how the different elements (mooring, buoys, receiver, signal flags, hydrophone, weights and ropes) are placed in relation to each other. On the right, a picture of a deployed receiver.

**Figure 7 sensors-22-08208-f007:**
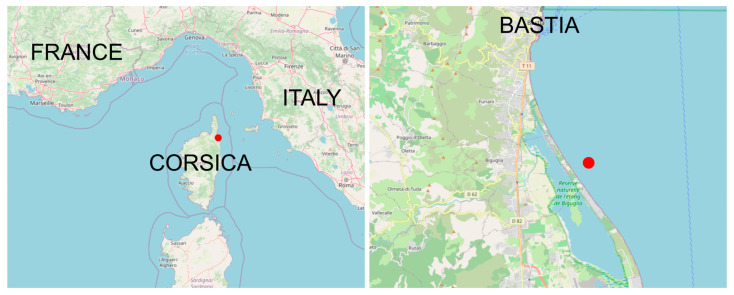
The experiment took place near the coastline south of the city of Bastia, in Corsica, a French Mediterranean island located between France and Italy. Red dot represents the location of the experiment.

**Figure 8 sensors-22-08208-f008:**
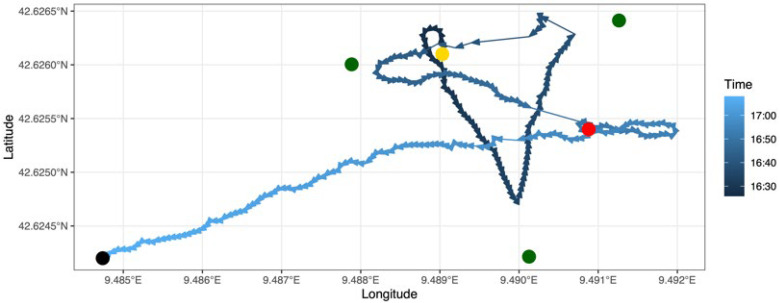
GPS track of the boat during the test experiment. The yellow point represents the starting point, and the black point represents the end point of the towed test. The red point represents the position from which the experiment for the determination of the maximum operational distance of our system began. Green points represent the receivers.

**Figure 9 sensors-22-08208-f009:**
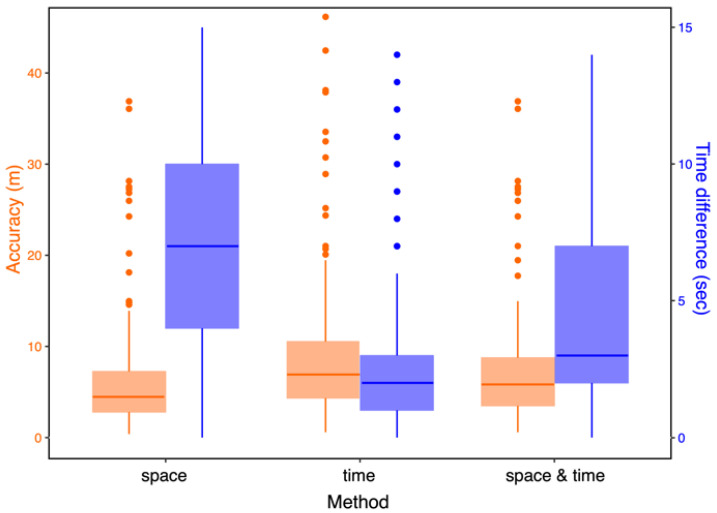
The performance of our system given for the three methods (space, time, space & time). For each method, the overall accuracy is available on the left in orange, while the time difference between the GPS positions of the boat and the reconstructed positions is available on the right in purple.

**Figure 10 sensors-22-08208-f010:**
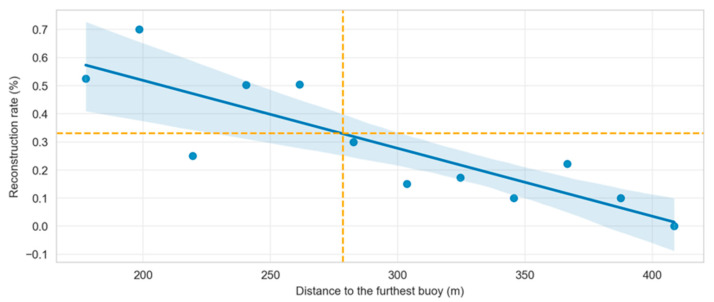
The reconstruction rate obtained in the context of our experiment with a maximum distance of 429.5 m from the furthest acoustic receiver. A linear regression was applied between 0 and 429.5 m to estimate the distance at which the 33% threshold is reached. Mean (blue line) and confidence intervals (light blue area) of the regression are displayed.

**Figure 11 sensors-22-08208-f011:**
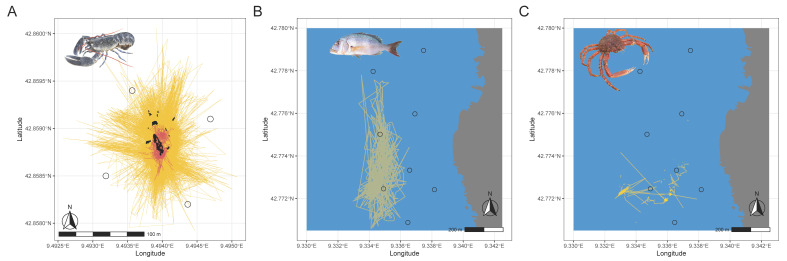
(**A**) A continuous-time correlated random walk model (CTCRW) is applied to raw TDOA data (yellow) to reveal more realistic movement paths (red) of a lobster tagged with a continuous pinger released at a wreck (black shape); (**B**) movement paths of an individual tagged dentex; (**C**) a spider crab in Nonza after applying the CTCRW model. The black circles correspond to the positions of the buoys of the real-time tracking system.

**Figure 12 sensors-22-08208-f012:**
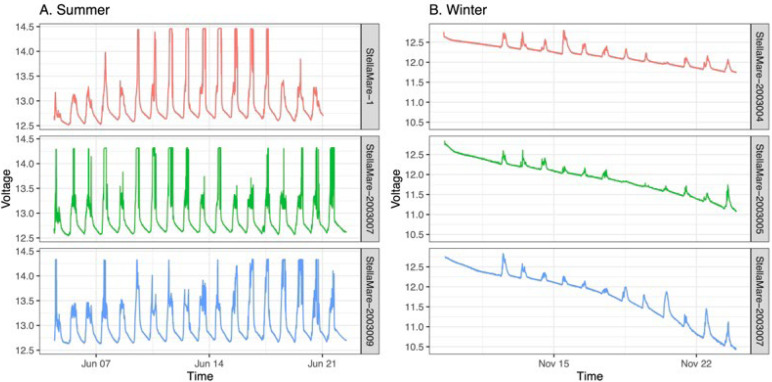
(**A**) Battery voltage plots of three acoustic receivers during summer experiments; (**B**) battery voltage plots of three acoustic receivers during winter experiments.

## Data Availability

All data files used for the performance analysis of our system are available in the following GitHub repository: https://github.com/fmmanicacci/sm_buoys.

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
