# Peer review of "A Wireless Autonomous Real-Time Underwater Acoustic Positioning System"

_sensors, 2022, doi:10.3390/s22218208_

Round 1

Reviewer 1 Report

Dear Authors, consider this recommendations:

- The problem description is poor

- Pictures are unclear

- Related works aren't presented

- Fig. 2 is no sense

- Fig. 3  has bad description

- Text needs extensively improvement

- Bad and unclear screenshot i Fig. 4

- The term TDOA is not defined and described

- I strongly recommend authors improve this manuscript. This way isn't possible to be minimally reviewed.

Author Response

Dear reviewer, thank you for your time. Please find our responses to all your comments below.

The problem description is poor.

We have updated the introduction. We hope to have improved the description of the problem.

Pictures are unclear.

We have updated the legend for Figures 2, 3 and 6 and hope that these will now be clearer.

Related works aren't presented.

We disagree. In the introduction we presented several systems (passive monitoring, VR2W, VRAP, JSTATS, MAP600 and LoRa based system named IoF) with their pros and cons. Also, we have provided in supplementary material a table with the performance of eight systems (HR-VPS, VR2AR, VR2W, WHS 3050 MAP, MAP 600, WHS 3050 MAP 76 and TBR 700) from the three major brands presents on the market (Vemco, Lotek and Thelma) collected in 9 different studies to compare our results. Also, the strengths and weaknesses of these systems have been compared to ours in the discussion section.

Fig. 2 is no sense

We introduced figure 2 to give a visual description of the acoustic receivers but also to convince that their shape and size are not an obstacle to its deployment at sea by a crew reduced to two people. Also, we wanted to show that the material had been properly protected against the ravages of the sea. We have updated the legend of Figure 2 and hope to have clarified its significance.

Fig. 3 has bad description

The description in figure 3 has been updated. We hope it will be of better quality than the previous one.

Text needs extensively improvement.

We are sorry to read that. As non-native speakers, we have done our best to produce an understandable and correct text. Without further guidance from you, we are not able to make improvements. However, we are willing to use the services offered by Sensors for verification, correction and improvement of the English if necessary.

Bad and unclear screenshot i Fig. 4

This screenshot contains all the information used for the theoretical validation of our choices in solar panel as it was specified in the text and legend. Each of the fields (label and value) are clearly readable. We do not agree on the fact that this one lacks clarity.

The term TDOA is not defined and described

We have added a description of the principles of the TDOA algorithm in section 2.3 Server and processing.

I strongly recommend authors improve this manuscript. This way isn't possible to be minimally reviewed.

We hope that the changes made to the manuscript will allow you to review properly this time.

Reviewer 2 Report

The paper is interesting and timely.

Section 2 general idea of positioning based on trilateration using a time difference algorithm is not described. It causes the paper to be not easy to understand for a typical reader. It should be added description of this algorithm in its section.

In the paper are a few typos and editing errors.

Author Response

Dear reviewer, thank you for your time. Please find our responses to all your comments below.

Section 2 general idea of positioning based on trilateration using a time difference algorithm is not described. It causes the paper to be not easy to understand for a typical reader. It should be added description of this algorithm in its section.

We have added a description of the principles of the TDOA algorithm in section 2.3 Server and processing.

In the paper are a few typos and editing errors.

We are sorry to read that. As non-native speakers, we have done our best to produce an understandable and correct text. We have tried to eliminate all typos and editing errors. However, if this is not enough, we are willing to use the services offered by Sensors for verification, correction and improvement of the English if necessary.

Reviewer 3 Report

In this paper, the authors a new high-resolution real-time acoustic tracking system and also provide the experimental results. The whole work looks solid and interesting. Some suggestions are presented as follows.

1. In Section 1, the authors present several published works. The authors are suggested to compare these work with the proposed system to highlight the main contributions in terms of complexity, cost, localization precision, and etc.

2. In section 2.4, the authors are suggested to provide detail about the geometry of the localization system including depth, horizontal range, and etc.

3. In Section 3, the results are presented, the authors are suggested to provide how to process the received raw data to get the results proposed in this paper.

Author Response

Dear reviewer, thank you your time. Please find our responses to all your comments below.

1. In Section 1, the authors present several published works. The authors are suggested to compare these work with the proposed system to highlight the main contributions in terms of complexity, cost, localization precision, and etc.

We have compared (performance, deployment, real time and network) in the discussion four (VR2W, JSTATS, MAP 600, IoF) of the seven systems presented in the introduction as we consider that they represent the most widely used or innovative systems currently available. The RATP, VRAP and HTI 290/291 systems have only been mentioned to give a broader overview of existing systems and their limitations. Also, please note that a comparative performance table of 8 systems is also included in the supplementary material.

2. In section 2.4, the authors are suggested to provide detail about the geometry of the localization system including depth, horizontal range, and etc.

We have reworked section 2.4 Installation procedure to bring more precision on the geometry of our geolocation system.

3. In Section 3, the results are presented, the authors are suggested to provide how to process the received raw data to get the results proposed in this paper.

The last part of Section 2.5 Performance tests (after Figure 8) explains our approach to estimating the accuracy of our system. We have updated this section so that it also explains how the reconstruction rate was obtained.
